# Agonist-antagonist muscle strain in the residual limb preserves motor control and perception after amputation

Hyungeun Song [1,2✉], Erica A. Israel[1], Samantha Gutierrez-Arango [1], Ashley C. Teng[1,3], Shriya S. Srinivasan [1,2], Lisa E. Freed [1] & Hugh M. Herr [1,4✉]

## Abstract

**Background** Elucidating underlying mechanisms in subject-specific motor control and perception after amputation could guide development of advanced surgical and neuroprosthetic technologies. In this study, relationships between preserved agonist-antagonist muscle strain within the residual limb and preserved motor control and perception capacity are investigated.

**Methods** Fourteen persons with unilateral transtibial amputations spanning a range of ages, etiologies, and surgical procedures underwent evaluations involving free-space mirrored motions of their lower limbs. Research has shown that varied motor control in biologically intact limbs is executed by the activation of muscle synergies. Here, we assess the naturalness of phantom joint motor control postamputation based on extracted muscle synergies and their activation profiles. Muscle synergy extraction, degree of agonist-antagonist muscle strain, and perception capacity are estimated from electromyography, ultrasonography, and goniometry, respectively.

**Results** Here, we show significant positive correlations ($P < 0.005$–$0.05$) between sensorimotor responses and residual limb agonist-antagonist muscle strain. Identified trends indicate that preserving even 20–26% of agonist-antagonist muscle strain within the residuum compared to a biologically intact limb is effective in preserving natural motor control postamputation, though preserving limb perception capacity requires more (61%) agonist-antagonist muscle strain preservation.

**Conclusions** The results suggest that agonist-antagonist muscle strain is a characteristic, readily ascertainable residual limb structural feature that can help explain variability in amputation outcome, and agonist-antagonist muscle strain preserving surgical amputation strategies are one way to enable more effective and biomimetic sensorimotor control postamputation.

**Plain language summary**

People who undergo limb amputation can have issues with controlling movement and perception of residual limbs. This, in turn, can impact the success of neuroprosthetic strategies, which use signals from the body to control a prosthetic limb. Here, we wanted to understand how sensory signals within the muscle help to preserve movement and limb perception following amputation. We used ultrasound imaging and other methods to measure muscle activity and limb perception in fourteen people who have undergone lower limb amputations. We show that the level at which the relationship between pairs of related muscles is preserved is associated with more natural control of limb movement after amputation. Developing surgical techniques that preserve this relationship may help people living with amputations to naturally perceive and control their residual limbs, and ultimately may improve controllability of assistive prosthetic devices.

[1] K. Lisa Yang Center for Bionics, Massachusetts Institute of Technology, Cambridge, MA, USA. [2] Harvard-MIT Division of Health Sciences and Technology, Massachusetts Institute of Technology, Cambridge, MA, USA. [3] Mechanical Engineering Department, Massachusetts Institute of Technology, Cambridge, MA, USA. [4] Harvard Medical School, Cambridge, MA, USA. ✉email: hngnsong@mit.edu; hherr@media.mit.edu

Proprioception is possible due to the presence of sensory organs within peripheral tissues including muscles, tendons, joint capsules, and skin[1,2]. Among these sensory organs, proprioception is primarily mediated by mechanoreceptors called muscle spindles and Golgi tendon organs which sense muscle length, speed, and tension[3]. Proprioceptive neural signaling relies on both microscale mechanotransduction processes[4], and macroscale biomechanically-functional tissue architectures[1]. The realization of such an architecture in a person with biologically intact limbs is implemented by mechanically-coupled antagonistic muscles spanning an articular joint that enables afferent signaling from the mechanoreceptors corresponding to limb movements through agonist-antagonist muscle strain (AMS).

The conventional standard-of-care amputation paradigm permanently disrupts the anatomical and neuromechanical principles of AMS, resulting in perturbed proprioception in people living with limb loss. Many ongoing efforts to restore locomotion for persons with leg amputation involve motor intent classification strategies based on electromyography (EMG) and intrinsic prosthetic signals[5,6]. However, in the absence of visual feedback, postural responses and balance during walking remain challenging for persons with leg amputation[7,8], which may indicate that motor control and proprioceptive percepts are significantly altered in persons that have undergone a conventional amputation procedure[9–11]. Toward better, more biomimetic control of an external prosthesis, invasive nerve interfacing using artificial electrical stimulation has shown great potential in restoring cutaneous[12–14] and proprioceptive sensation[11,12]. However, due to the complexity of afferent signaling through artificial nerve stimulation, and the relatively limited resolution of state-of-the-art implantable devices, it can be challenging to engineer stable neuroprosthetic interfaces that offer natural cutaneous and proprioceptive percepts.

As an alternative approach to neuroprosthetic interface design, surgical methodologies to reconfigure residual limb soft tissues may lower the burden of engineering and offer a more efficacious, biomimetic motor control strategy while also providing feedback from the amputated limb via biological sensory organs[15]. Conventional myocutaneous flap amputation procedures prioritize creating enough muscle padding for prosthetic socket fitting[16]. To enhance neuroprosthetic control, amputation paradigms seeking further reconfiguration of residual limb soft tissues have been developed, including Targeted Muscle Reinnervation (TMR)[17–20], Regenerative Peripheral Nerve Interfaces (rPNIs)[21–23], and the Agonist-antagonist Myoneural Interface (AMI)[24–27]. Each of these techniques have been demonstrated in combination with neuroprostheses[17,19,22,24] for the enhancement of prosthetic control. Nevertheless, neuroprosthetic performance is a system-level evaluation that depends on multiple factors such as subject-specific inherent capacities of residual limb motor control and phantom limb perception, engineered functional feedback, presence of visual feedback, and choice of the neuroprosthetic control paradigm. Consequently, there is a clear and present need to uncover the fundamental nature of how surgical residual limb reconstruction alone impacts clinical outcomes after amputation.

In this study, we investigate motor control and phantom limb perception capacities of 14 clinical research subjects having unilateral transtibial amputation spanning a range of ages and etiologies. Of the 14 participants, 7 subjects had received the AMI amputation, and 7 had undergone a non-AMI amputation. The transtibial AMI amputation comprises the surgical creation of dynamic agonist-antagonist muscle pairs for the enhancement of AMS. One muscle pair is constructed for the ankle joint comprising the lateral gastrocnemius linked to the tibialis anterior, and a second muscle pair for the subtalar joint comprising the peroneus longus linked to the tibialis posterior[22]. The study focuses on the impact of AMS preservation within the residual limb on motor control and phantom limb perception. We hypothesize that enhanced levels of residual limb AMS will improve motor control naturalness and proprioceptive perception postamputation in persons with transtibial amputation. To evaluate this hypothesis, the study clinically evaluates the naturalness of motor control and limb perception capacity during ankle and subtalar joint movements without visual or any other functional feedback. We collected muscle electromyography patterns and two degrees-of-freedom (2-DoF) kinetic data during bilateral, mirrored movements between the intact and phantom ankle-foot limbs of each subject. For these mirrored movements, we assessed the degree of residual limb AMS using ultrasonography. Because motor control in biologically-intact limbs is executed by the activation of combinations of muscle synergies, we evaluate motor control naturalness and limb perception of amputees using muscle synergy analysis[28–30]. The study findings support the hypothesis that enhancing AMS in the residual limb improves motor control naturalness and perception after amputation, underscoring the importance of surgical techniques such as the AMI that create a residuum tissue structure that preserves agonist-antagonist muscle dynamics.

## Methods

**Study design and clinical evaluation.** The present work begins to investigate one outcome measure—economy of motion—of our ongoing clinical trial, NCT03913273, although we do not report any pre-specified endpoints of that trial in the present work. In overall scope the NCT03913273 trial investigates if AMIs can (i) improve voluntary free-space prosthetic control, (ii) improve voluntary and involuntary (reflexive) prosthetic terrain adaptations, and (iii) serve as a bidirectional human-device interface after transtibial amputation (https://clinicaltrials.gov/). The relationship of the present work to that trial is to obtain preliminary data and an algorithmic framework—a muscle synergy model—and thus inform our assessment of the pre-specified outcome measures for NCT03913273. Computational tools such as the model investigated in the present work are a critical part of studying and quantifying voluntary motor control postamputation in free space, ambulatory ascent and descent of stepping stairs while wearing a prototype multi-degree-of-freedom prosthesis, and potential for closed-loop prosthetic control by functional electrical stimulation.

All data in the present study, were collected at the Massachusetts Institute of Technology (MIT) under IRB approval from our Committee on the Use of Humans as Experimental Subjects (protocol 1812634918). All participants signed informed consent forms prior to data collection. The work followed the same prospective, non-randomized study design described in NCT03913273. The time period of recruitment and data collection was June 12, 2019 through September 19, 2021. Eligibility criteria included transtibial amputee subjects within an age range of 18 to 65 years, a fully healed amputation site, proficiency in the use of a standard lower-extremity prosthesis, and capability for ambulation with variable cadence (K level 3 and 4[31]). Exclusion criteria included one more of the following underlying health conditions: cardiopulmonary instability manifest as coronary artery disease, chronic obstructive pulmonary disease, extensive microvascular compromise, as well as persons who are pregnant and/or active smokers.

Table 1 of the present work summarizes the 14 study participants, listing a total of 7 AMI subjects and 7 non-AMI control (CTL) subjects. The age of the subjects ranged from 25 to 62 years. The male to female ratio was 5:2. The subjects represented different amputation types: AMI[25] (7/14), conventional[16] (6/14), and Ertl osteomyoplasty[32,33] (1/14). AMI amputation surgeries

**Table 1 Study population.**

| Participant ID | Amputation type | Age (years) | Time since amputation (years) | Amputation etiology | Biological sex | Height (m) | Weight (kg) |
|---|---|---|---|---|---|---|---|
| AMI-1/BIO-A1 | AMI | 43 | 1.6 | Thermal Injury | Female | 1.68 | 81 |
| AMI-2/BIO-A2 | AMI | 55 | 2.7 | Trauma | Male | 1.73 | 77 |
| AMI-3/BIO-A3 | AMI | 50 | 1.0 | Trauma | Female | 1.68 | 81 |
| AMI-4/BIO-A4 | AMI | 58 | 1.2 | Trauma | Male | 1.90 | 93 |
| AMI-5/BIO-A5 | AMI | 32 | 0.5 | Trauma | Male | 1.75 | 75 |
| AMI-6/BIO-A6 | AMI | 29 | 0.6 | Trauma | Male | 1.68 | 84 |
| AMI-7/BIO-A7 | AMI | 48 | 0.5 | Trauma | Male | 1.70 | 75 |
| CTL-1/BIO-C1 | Standard | 25 | 1.4 | Oncological | Female | 1.64 | 54 |
| CTL-2/BIO-C2 | Standard | 62 | 2.7 | Trauma | Female | 1.65 | 81 |
| CTL-3/BIO-C3 | Standard | 25 | 2.0 | Talipes Equinovarus | Male | 1.78 | 108 |
| CTL-4/BIO-C4 | Standard | 39 | 2.7 | Trauma | Male | 1.60 | 63 |
| CTL-5/BIO-C5 | Ertl osteo-myoplasty | 62 | 2.6 | Trauma | Male | 1.80 | 97 |
| CTL-6/BIO-C6 | Standard | 61 | 5.3 | Trauma | Male | 1.73 | 91 |
| CTL-7/BIO-C7 | Standard | 46 | 8.7 | Trauma | Male | 1.78 | 75 |
| Mean ± s.d. | | 45 ± 14 | 2.4 ± 2.2 | | | 1.72 ± 0.08 | 81 ± 14 |

*AMI* Agonist-antagonist Myoneural Interface, residual limbs of participants who underwent an AMI amputation (AMI-1-7), residual limbs of participants who underwent a Non-AMI control amputation (CTL-1-7), unaffected biologically-intact limbs (BIO-A1-7 and BIO-C1-7).

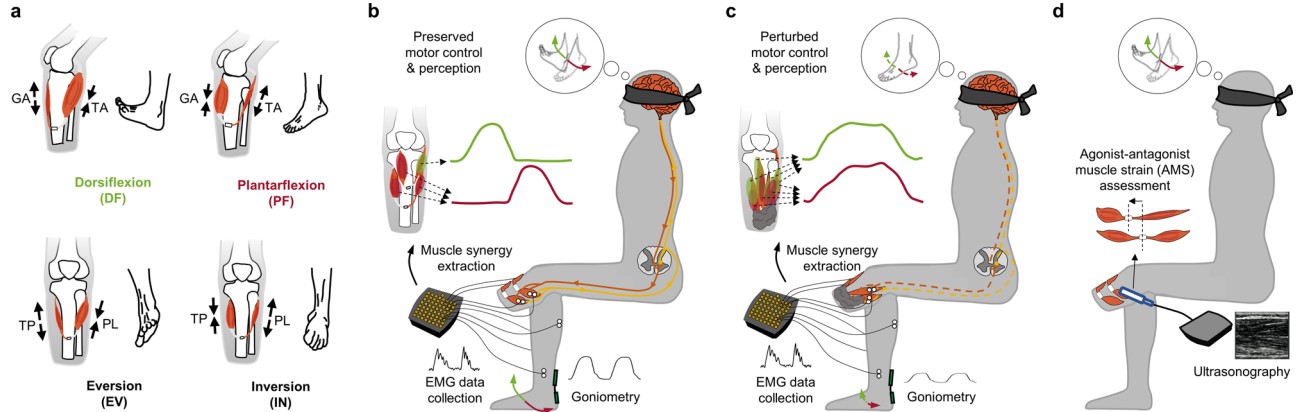

**Fig. 1 Clinical evaluation of sensorimotor responses and the degree of agonist-antagonist muscle strain (AMS) for participant's residual limb muscles.** Shown in **a**, the Agonist-antagonist Myoneural Interface (AMI) amputation seeks to emulate physiological actuation of antagonistic muscle contraction and stretch. Ankle and subtalar AMI constructs are devised to create direct agonist-antagonist coupling for ankle dorsi and plantarflexion and for subtalar eversion and inversion. For the ankle joint AMI construct, the tibialis anterior (TA) is linked to the lateral gastrocnemius (GA), and for the subtalar joint AMI construct, the tibialis posterior (TP) is linked to the peroneus longus (PL). In **b** and **c**, the experimental setup is shown. Motor control and phantom limb perception capacity are assessed in free space without visual or any other functional feedback. Perturbed motor control and perception are anticipated if a critical degree of AMS is not preserved in the limb. Representations of the dorsi and plantarflexion synergic motor outputs are shown in green and red, respectively; efferent and afferent neural signals are shown in brown and yellow, respectively. Eversion and inversion were also tested but are not shown here. In **d**, AMS is computed from muscle fascicle changes during cyclic phantom ankle and subtalar joint movements. Here, the size and positioning of elements are representative and not to scale.

had been done according to Partner's Institutional Review Board protocol p2014001379 as in previous reports[24,25].

As noted in the introduction section, the AMI transtibial amputation procedure creates mechanical linkages between two pairs of natively vascularized and innervated muscles within the residual limb; one pair for the missing ankle joint and another pair for the missing subtalar joint[22,23]. For the ankle joint AMI construct, the tibialis anterior (TA) was linked to the lateral gastrocnemius (GA), and for the subtalar joint AMI construct, the tibialis posterior (TP) was linked to the peroneus longus (PL). The AMI amputation aims to emulate physiological antagonistic actuation between the residual limb muscles to restore AMS (Fig. 1a). In distinction, some non-AMI amputations may perturb AMS by severing or restricting residual agonist-antagonist muscle movements. Agonist-antagonist muscle couplings were not specifically reconstructed during conventional amputation or

Ertl osteomyoplasty amputation procedures[16,32,33]. Thus, the study population represented differing degrees of AMS within the residual musculature.

We collected EMG simultaneously from the TA, GA, TP, and PL muscles of both the residual limbs (AMI, CTL) and unaffected biologically intact limbs (BIO-A, BIO-C). A 2-DoF goniometer was also placed on the posterior aspect of the unaffected BIO limb spanning the ankle-foot complex (Supplementary Fig. 1a–c) to record mirrored movements between the intact and perceived phantom limb. We provided multiple motor control task instructions via on-screen and audio recordings. During motor control trials, no visual or other functional feedback was provided in an effort to focus on investigating the impact of proprioceptive feedback on motor control and limb perception capacity (Fig. 1b, c). To compute AMS, we utilized ultrasonography to record from the residual limbs while each subject repeated cyclic

plantarflexion-dorsiflexion (PF-DF) and inversion-eversion (IN-EV) mirrored phantom limb movements (Fig. 1d, Supplementary Fig. 1d, e). The maximum muscle fascicle strains were then estimated from ultrasound video recordings which were further normalized by the nominal muscle fascicle strain ranges from a computational musculoskeletal limb model[34]. The average PF-DF and IN-EV AMS values were utilized to represent the degree of AMS within the residuum. Given this definition, the degree of AMS ranges from 0 to 1, where zero indicates that the subject preserved none of the AMS present in the biologically intact limb, and 1 indicates fully preserved biological AMS.

**Surface electrodes placements and EMG processing**. Bipolar surface electrodes were acutely placed over each of the target muscles for EMG recording. The target muscles include GA, TA, TP, and PL of both the residuum and unaffected limb (Supplementary Fig. 1a). Ultrasound imaging was used to guide electrode placement when it was needed. All the EMG signals were off-line high-pass filtered (fourth-order zero-lag Butterworth filter, 20 Hz cut-off frequency). The filtered EMG signals were full-wave rectified and low-pass filtered (fourth-order zero-lag Butterworth filter, 5 Hz cut-off frequency) to compute muscle activation patterns. All EMG signals were normalized to calibrated maxima for each muscle.

**Muscle synergy extraction and synergy activation profile**. The motor control of residual limbs was evaluated by muscle synergy extraction from muscle activation patterns during discrete ankle and subtalar joint movement trials. Subjects were asked to sequentially make PF, IN, DF, and EV movements of both the residual (phantom) and biologically intact limbs. The order of discrete movement trials was not randomized in an effort to identify existing motor capabilities for the 4 principal movements as accurately as possible. Discrete movements were repeated 40 times. We used a generally accepted mathematical model[29] for the representation of motor outputs as muscle synergy combinations as

$$\boldsymbol{m}(t) = \sum_{i=1}^{N} c_i(t)\boldsymbol{w}_i + \varepsilon(t) \tag{1}$$

where $\boldsymbol{m}(t)$ is the muscle activation patterns at time $t$; $\boldsymbol{w}_i$ is the $i$th muscle synergy vector; $c_i(t)$ is the time-varying coefficient, or synergy activation, for $i$-th muscle synergy vector; $N$ is the total number of muscle synergy vectors composing the muscle activation patterns; and $\varepsilon(t)$ is the residual. Briefly, this model represents the muscle activation patterns as linear combinations of a set of time-invariant muscle synergy vectors that are activated by time-varying coefficients. As is generally accepted in the field, we consider $\boldsymbol{w}_i$ as a muscle synergy profile that has a structural basis in the nervous system and consider $c_i(t)$ as an index of motor commands, or synergy activations. Muscle synergies were extracted by the non-negative matrix factorization (NMF) algorithm[35]. The NMF was started with the initialization of time-varying coefficients and muscle synergy vectors to random positive values in the [0 1] interval. The goodness of fit metric of decomposed matrices was evaluated by variance accounted for (VAF)[36]. The NMF was continued until the change in VAF in 50 consecutive iterations was less than a tolerance of $1 \times 10^{-5}$. To reduce the probability of finding a local minimum solution for the NMF optimization, the same procedures were repeated 30 times with different sets of initial conditions, and the solution with the most VAF was selected. The number of muscle synergies was selected as the least number of synergies that could adequately reconstruct the muscle activation patterns, as determined by VAF > 0.95[36]. To enable intrasubject comparisons of motor

commands, the average vectors of synergy activation profiles $\boldsymbol{u}_{\text{task}}$ were used, or

$$\boldsymbol{u}_{\text{task}} = \sum_{i=1}^{N} \frac{\int_0^{T_{\text{task}}} c_i(t)dt}{T_{\text{task}}} \hat{\boldsymbol{n}}_i \tag{2}$$

where $\hat{\boldsymbol{n}}_i$ is a unit vector in synergy space indicating activation of synergy vector $\boldsymbol{w}_i$; $T_{\text{task}}$ is a time period of a given task. To identify motor commands for PF, DF, IN, and EV, the muscle activation patterns for each discrete movement were gathered and synergy activation vectors were computed independently. Then, the synergy activation vectors were normalized for further analysis.

**Naturalness of muscle synergy and synergy activation**. We quantified the naturalness of muscle synergy and synergy activation profiles of AMI and CTL groups by computing their similarities to the average normalized values of the BIO group. Specifically, a muscle synergy of one subject was considered to correspond to a muscle synergy of another subject when the maximum of the scalar products was found among others. After sorting the muscle synergy vectors, the representative muscle synergy vectors of the BIO group were determined as the normalized average muscle synergy vectors of the BIO group. Finally, the naturalness of one's muscle synergy was calculated by plotting the mean scalar products with the representative of the BIO group. Similarly, the naturalness of one's synergy activation vectors was calculated by plotting the mean scalar products with the normalized average synergy activation vectors of the BIO group. For the BIO group, a leave-one-out procedure was used for computing their dot products. The universal number of synergy vectors for similarities analysis was unified to 3 muscle synergy vectors, which was the number of synergy vectors of all subjects in the BIO group. When fewer synergy vectors were identified previously by synergy extraction procedures, $\boldsymbol{0}$ vectors were added to muscle synergy and synergy activation vectors to match the dimensionality of vectors.

**Robustness of synergy activation**. We quantified the robustness of ankle and subtalar volitional control based on the degree of decoupling between synergy activations for different target movements. The angle between two average vectors of synergy activations for two different target movements indicates the tolerance to variance in motor commands of two corresponding targeted movements. When the tolerance of motor commands is larger than the expected variance, it indicates that a subject can reliably produce distinguishable synergy activation for two discrete movements of interest. Therefore, the margin of synergy activation is given as

$$\varnothing_{ij} = \text{acos}\left(\boldsymbol{u}_i^T \bullet \boldsymbol{u}_j\right) - \frac{1}{T_{\text{task},i}} \int_0^{T_{\text{task},i}} \left| \text{acos}\left(\boldsymbol{u}_i^T \bullet \frac{\boldsymbol{c}_i(t)}{\parallel \boldsymbol{c}_i(t) \parallel}\right) \right| dt$$
$$- \frac{1}{T_{\text{task},j}} \int_0^{T_{\text{task},j}} \left| \text{acos}\left(\boldsymbol{u}_j^T \bullet \frac{\boldsymbol{c}_j(t)}{\parallel \boldsymbol{c}_j(t) \parallel}\right) \right| dt$$
$$(i \neq j, i, j \in \{\text{PF}, \text{DF}, \text{IN}, \text{EV}\}) \tag{3}$$

where $\phi_{ij}$ is the margin in synergy activations to have distinguishable patterns between targeted $i$ and $j$ discrete movements; $T_{task,i}$ is the interval time of $i$ discrete movement. Note that the first term in the right-hand side of the equation is the angle between two synergy activation average vectors (tolerance); the second and third terms are variability in motor commands corresponding to the two discrete movements. The thresholds of 0 were selected for synergy activation margin to determine robustly decoupled discrete movements.

**Synergy space ($U_s$-space) and motor intent decoding ($\alpha$-space).** We decoded the motor intents from arbitrary muscle activation patterns based on the extracted muscle synergy and synergy activation average vector of the 4 principal movements (PF, DF, IN, EV). The time-varying coefficients of the arbitrary muscle activation patterns were decomposed by revised NMF, fixing synergy vectors as the extracted muscle synergy from discrete movement trials during NMF iterations[37]. The same initialization and iteration protocols were utilized as those of muscle synergy extraction procedures for the rest of decomposition procedures. Given the revised NMF, the arbitrary muscle activation patterns can be reflected into the muscle synergy space of the discrete ankle and subtalar joint movements. This reflected time-varying coefficients at time $t$, $U_S(t)$, is further decoded into motor intents of ankle and subtalar movements by using the synergy activation vector of 4 discrete movements, $\boldsymbol{u}_{PF}$, $\boldsymbol{u}_{DF}$, $\boldsymbol{u}_{IN}$, and $\boldsymbol{u}_{EV}$, as

$$\alpha_{PFDF}(t) = \frac{1}{\mathrm{acos}(\boldsymbol{u}_{PF}^T \bullet \boldsymbol{u}_{DF})} \left( \mathrm{acos}\left( \boldsymbol{u}_{PF}^T \bullet \frac{U_S(t)}{\| U_S(t) \|} \right) - \mathrm{acos}\left( \boldsymbol{u}_{DF}^T \bullet \frac{U_S(t)}{\| U_S(t) \|} \right) \right) \quad (4)$$

$$\alpha_{INEV}(t) = \frac{1}{\mathrm{acos}(\boldsymbol{u}_{IN}^T \bullet \boldsymbol{u}_{EV})} \left( \mathrm{acos}\left( \boldsymbol{u}_{IN}^T \bullet \frac{U_S(t)}{\| U_S(t) \|} \right) - \mathrm{acos}\left( \boldsymbol{u}_{EV}^T \bullet \frac{U_S(t)}{\| U_S(t) \|} \right) \right) \quad (5)$$

where $\alpha_{PFDF}$ and $\alpha_{INEV}$ indicate directions of desired movements in ankle and subtalar DoF, respectively. Given these definitions, $\alpha_{INEV}$ and $\alpha_{PFDF}$ ranges from $-1$ to 1 and the $\alpha$-space consists of $\alpha_{INEV}$ and $\alpha_{PFDF}$ served as a phase domain of motor control. The universal dimensionality of decomposed $U_S$ was unified as 3, which was the dimensionality of decomposed $U_S$ of all subjects in the BIO group. When fewer synergy vectors were identified previously by synergy extraction procedures, $\boldsymbol{0}$ vectors were added in $U_S$ to match dimensionality of vectors.

**2-DoF motor controllability.** We quantified simultaneous multi-DoF motor controllability by investigating the transitions in directionality of motor intent during 10 cycles of the drawing-a-circle tasks. First, the $U_S$ was reflected into $\alpha$-space. To draw an ideal circle in joint space, the directionality of motor intent in both ankle and subtalar DoF needs to be changed simultaneously. This is equivalent to simultaneous changes in both $\alpha_{INEV}$ and $\alpha_{PFDF}$, resulting in diagonal trajectory in $\alpha$-space. Meanwhile, if the subject is only able to perform a single DoF motor control at a time, only changes in $\alpha_{INEV}$ or $\alpha_{PFDF}$ is found at a time. This is equivalent to a horizontal or vertical trajectory in $\alpha$-space. Therefore, 2-DoF motor controllability was calculated by integrating diagonal components of trajectories within $\alpha$-space to evaluate the simultaneous multi-DoF motor controllability, or

$$\text{2-DoF motor controllability} = 1 - \frac{1}{\pi} \sum_{j=1}^{4} \sum_{i=1}^{n_j} \frac{1}{T_{E,i}^j - T_{S,i}^j} \int_{T_{S,i}^j}^{T_{E,i}^j} \mathrm{acos}\left( \delta_{11} \cdot \frac{|\dot{\alpha}|}{\|\dot{\alpha}\|} \right) dt \quad (6)$$

where $n_j$ is the number of trajectories in $j$-th quadrant of $\alpha$-space; $T_{S,i}^j$ and $T_{E,i}^j$ indicate the start and end time of $i$th trajectory in $j$th quadrant, respectively; $\dot{\alpha}$ and $\delta_{11}$ are a velocity vector in $\alpha$-space and unit diagonal vector, given as $[\frac{1}{\sqrt{2}}, \frac{1}{\sqrt{2}}]$, respectively. The trajectories in $\alpha$-space were evaluated by each quadrant independently to investigate multi-DoF motor controllability of different combinations of discrete movements. The average diagonal

components of trajectories on each quadrant were computed and normalized by $\frac{\pi}{4}$. Finally, the mean values of diagonal components of all quadrants were calculated. If no full trajectories were present in a quadrant, it was considered as zero diagonal component for that quadrant. Note that the second term on the right-hand side of the equation converges to zero and 2-DoF motor controllability becomes 1 when all the trajectories in all quadrants are composed by only diagonal components. Thus, given this definition, 2-DoF motor controllability ranges from 0 to 1.

**Evaluation of spatiotemporal motor control under time constraints.** Spatiotemporal motor control was evaluated from two metrics, motor control performance and economy of motion under increasing time constraints from 2.0 s to 1.5 s, 1 s, 0.8 s, and 0.5 s. For the visualization, an index of difficulty (ID) of speed-accuracy task for each time constraint (2.0–0.5 s) was calculated as the logarithm of the inverse of the time constraint and scaled to range from 0 to 1 ($ID_{2.0}$-$ID_{0.5}$), inspired by Fitts' law[38–40]. Speed-accuracy tasks comprised 10 repetitions in each of discrete PF, DF, IN, EV movements in a random order for each of 5 time-interval settings. The motor control performance was analyzed based on the tracking errors between the decoded motor intent $\boldsymbol{\alpha}$, and ideal targets of $j$ discrete movement $\chi_j$ in $\alpha$-space, as

$$\text{motor control performance} = 1 - \frac{1}{2(T_E - T_S)} \int_{T_S}^{T_E} |\chi_j - \alpha(t)| dt. \quad (7)$$

Ideal target movements in $\alpha$-space $\chi_{PF}$, $\chi_{DF}$, $\chi_{IN}$, and $\chi_{EV}$ were defined as $(0, -1)$, $(0, 1)$, $(-1, 0)$, and $(1, 0)$, respectively. Given this definition, the motor control performance shows how one can maintain motor intent corresponding to given motor tasks. Economy of motion was computed by the ratio of effective synergy activation for targeted $j$ discrete movements to total synergy activation, or

$$\text{economy of motion} = \frac{1}{T_E - T_S} \int_{T_S}^{T_E} \frac{|\alpha_j(t)|}{\|\alpha(t)\|} dt \quad (8)$$

where $\alpha_j$ is effective synergy activation for targeted $j$ discrete movements, determined as $\alpha_{PFDF}$ for PF and DF and $\alpha_{INEV}$ for IN and EV. Given this definition, the economy of motion indicates the trajectory straightness of movements that were produced to achieve the target discrete movements.

**Assessment of phantom limb perception capacity.** A psychometric task was used to assess limb perception capacity of the phantom limbs across the full perceived ranges of motion (ROM) for DF-PF and IN-EV. The mirrored perceived phantom limb positions were measured by goniometry from the subjects' BIO limb and were normalized by the ROM of the BIO limb ($\hat{\theta}$) to allow comparison when plotted against the intended phantom limb positions ($\theta$) assessed from the EMG data. The intended limb position was assessed by the average value of $U_S \parallel \alpha_{PFDF}$ and $U_S \parallel \alpha_{INEV}$ for each movement which show the both direction and amplitude of desired movements. To vary the phantom limb position while considering the range of motion of each joint, the subjects were guided to perform 25, 50, 75, and 100% PF and DF range of motion during separate, 40 randomized PF and DF trials, and 50 and 100% range of motion during 30 randomized IN and EV trials. The limb perception capacity was estimated based on the relationship between the intended phantom limb positions and the mirrored perceived phantom limb positions, determined as the mean value of the ranges between 5 and 95% of the psychometric functions. When a subject reported either zero phantom limb sensation or inconsistent phantom limb sensation to

the intended limb positions, the limb perception capacity was determined as zero. When 5 or 95% of the psychometric function was not reached, the minimum or maximum value of the psychometric function was selected.

**Assessment of phantom limb sensations**. The reported phantom limb sensation scores focused on the vividness of their phantom limb sensations compared to actual sensations of their biologically intact limb during ankle, subtalar, and ankle/subtalar joint rotations. Subjects self-reported vividness of these phantom sensations on a scale of 0-to-10, where values of 0 and 10 respectively indicated no sensation or equivalent sensations for their phantom joint to their BIO limb. The full findings are given in Supplementary Table 1.

**Statistics**. No statistical methods were used to predetermine sample size a priori, but effect sizes were determined for the main outcomes using Cohen's $d$ values between 1.26-to-1.35 (Supplementary Table 2). Data collection and analysis were not performed blind to the conditions of the experiments. The unaffected limbs of all subjects served as the biologically intact limb population when appropriate, thus no separate non-amputated subjects were recruited. In all experiments, except when specifically noted, the order of the motor control tasks was randomized as described in the relevant Methods sections and in the Nature Research Reporting Summary. Six sensory-motor response variables were correlated with the degree of AMS for the pooled dataset of all 14 subjects' residual limbs (CTL-1-7 and AMI-1-7). A nonparametric correlation, Kendall's tau ($\tau$), and $P$ value were computed to address a positive association between each response variable and the degree of AMS. A first order exponential response curve was fitted to address a critical degree of AMS ($AMS_c$) that preserves 95% of each sensory-motor response variable. Jackknife mean ± standard deviation (s.d.) of $AMS_c$ and $R^2$ values of fitted response curve for each response variable were reported. The normality of the motor control data was tested by a Shapiro-Wilk test at a significance level of $\alpha = 0.05$. To consider within-subject limb differences (AMI:BIO-A and CTL:BIO-C), paired one-tailed $t$-tests were used at a significance level of $\alpha = 0.05$, as all motor control data did not violate the data normality. To consider between-subject residual limb differences (AMI:CTL), unpaired two-tailed $t$-tests were used at a significance level of $\alpha = 0.05$. Interactive effects between limb subgroups (AMI:CTL x affected:unaffected limb) were analyzed by 2-way ANOVA at a significance level of $\alpha = 0.05$. The full statistics are reported in the Supplementary Table 2.

**Reporting summary**. Further information on research design is available in the Nature Research Reporting Summary linked to this article.

## Results
**Naturalness of motor control**. Varied motor control in biologically intact limbs is executed by the activation of combinations of muscle synergies (Fig. 2a)[28–30]. We evaluated the naturalness of motor control of amputees by computing similarities between muscle synergies and their activation profiles for residual limbs and biologically intact limbs. We used a linear synergy model to extract muscle synergy and activation profiles from recorded EMG using a dimensionality-reduction technique[28–30]. The subjects performed discrete ankle and subtalar motion tasks that were assigned in the chronological order of PF-IN-DF-EV, which was not randomized in an effort to identify existing muscle synergies for the 4 principal movements as accurately as possible for the forthcoming analyses. The extracted muscle synergies and

average vectors of synergy activation profiles for all 28 limbs are shown in Fig. 2b, c, respectively. All BIO-A, BIO-C, AMI, and 4/7 CTL limbs indicated 3 muscle synergies performing 4 principal ankle and subtalar movements. All BIO and AMI limbs shared common muscle synergies; one synergy ($W_1$) was dominated by GA and TP muscle activations, and the other 2 synergies ($W_2$ and $W_3$) were respectively dominated by PL and TA activations. However, muscle synergies were altered in 5/7 CTL limbs as follows: CTL-1-4 all differed from BIO limbs in $W_2$ or $W_3$, showing coactivation tendencies. CTL-2-4, and CTL-6 also differed from BIO limbs in $W_1$, showing altered GA and TP coordination profiles (Supplementary Fig. 2b).

The muscle synergy similarity (m.s.s.) and synergy activation similarity (s.a.s.) between each subject's residual limb and the average profiles across all 14 subjects' biologically intact limbs was plotted against the degree of AMS (Fig. 2d, e). Both trends showed significant positive associations (m.s.s.: $\tau = 0.54$, $P < 0.01$; s.a.s.: $\tau = 0.56$, $P < 0.005$). Relatively low values of the critical degree of AMS ($AMS_c$) were anticipated to preserve 95% of natural discrete motor control for the residual limb (m.s.s.: $AMS_c = 0.21$, $R^2 = 0.94$; s.a.s.: $AMS_c = 0.22$, $R^2 = 0.84$). The results provide evidence in support of the degree of AMS within residual muscles enabling natural, discrete motor control.

Further analyses (Fig. 2f, g, and Supplementary Table 2) found significant differences for AMI:CTL comparisons (m.s.s.: $t = 3.30$, $P < 0.01$, s.a.s.: $t = 2.99$, $P < 0.02$), significant differences for CTL:BIO-C comparisons (m.s.s.: $t = 3.37$, $P < 0.01$, s.a.s.: $t = 3.27$, $P < 0.01$), and no significant differences for AMI:BIO-A comparisons (m.s.s.: $t = 0.48$, $P = 0.33$, s.a.s.: $t = 1.79$, $P = 0.06$). Further, subgroup analyses amongst the 28 limbs revealed significant interactive effects (AMI:CTL × affected:unaffected limb) for muscle synergy similarity and synergy activation similarity (m.s.s: $F = 10.68$, $P < 0.005$, s.a.s.: $F = 9.22$, $P < 0.01$). Together, the results suggest that advanced amputation procedures that actively preserve even a small degree of AMS may effectively preserve natural discrete motor control after amputation.

**Robust multi-degrees-of-freedom motor control**. High reliability of motor control that is consistent and cohesive, even in the absence of visual or other functional feedback, is anticipated to provide stable neural signaling and thereby enable neuroprosthetic control. To assess robust motor control for ankle and subtalar joint movements, we investigated the variability in synergy activation profiles for each movement (Fig. 3a). Synergy activation profiles differed for different movement tasks. All BIO-A, BIO-C, and AMI limbs produced 4 distinct synergy activations for the ankle and subtalar joint movements (Fig. 3b, Supplementary Fig. 2b). In contrast, only 3/7 CTL limbs were able to produce 4 distinct synergy activations for ankle and subtalar movements.

To evaluate simultaneous multi-DoF motor controllability, we asked the subjects to attempt 'drawing a circle' with mirroring, of their phantom foot and biologically intact foot by simultaneously controlling their ankle and subtalar joints. We decoded the measured muscle activations into synergy activations (Fig. 3c, $U_S$) using the previously identified muscle synergies ($W_1$-$W_3$) and a matrix decomposition technique[37]. We then computed motor intents from the synergy activations based on the average vectors of synergy activation profiles of the 4 principal ankle and subtalar joint movements, transforming into the $\alpha$-space (Fig. 3c, Supplementary Fig. 3). The $\alpha$-space presents the directions of desired movements from the decoded synergy activations. When the circle is drawn with ankle and subtalar movement transitions in concert, the directionality of motor intents for ankle and subtalar joints changes simultaneously, resulting in diamond-type trajectories consisting of diagonals in $\alpha$-space (Fig. 3d, e). In contrast, single

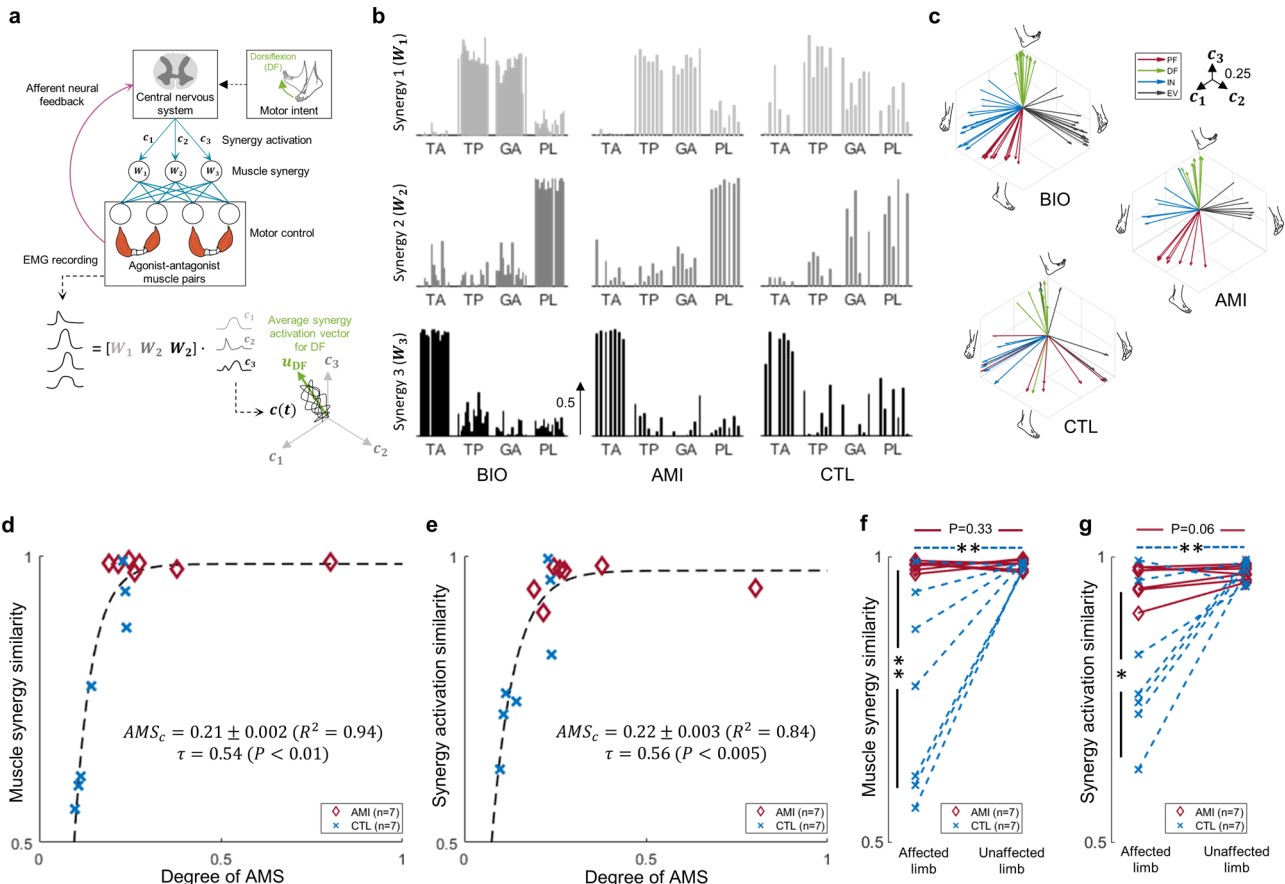

**Fig. 2 Naturalness of motor control.** In **a**, the framework of muscle synergies is shown. Varied motor control is executed by the activation of combinations of muscle synergies. In **b**, the extracted muscle synergies of all 28 limbs including the affected limbs of 14 subjects who had undergone either an Agonist-antagonist Myoneural Interface (AMI) amputation or a non-AMI amputation (CTL), and their unaffected limbs (BIO) are shown (BIO: $n = 14$, AMI: $n = 7$, CTL: $n = 7$). The muscle synergies were extracted from motor outputs of 4 muscles: the tibialis anterior (TA), tibialis posterior (TP), lateral gastrocnemius (GA), and peroneus longus (PL). In **c**, the average vectors of synergy activation profiles are shown. These vectors were derived from the muscle synergies found during discrete motion testing of all 28 limbs (BIO: $n = 14$, AMI: $n = 7$, CTL: $n = 7$). Each axis ($c_1$-$c_3$) indicates the activation of a muscle synergy ($W_1$-$W_3$). Here, the four different colors indicate the movements of plantarflexion (PF, red), dorsiflexion (DF, green), inversion (IN, blue), and eversion (EV, black). In **d** and **e**, the relationships between muscle synergy similarity and synergy activation similarity are shown with the degree of agonist-antagonist muscle strain (AMS) in a combined analysis of affected AMI and CTL limbs ($n = 14$). Reported are the Kendall's tau ($\tau$), P, $R^2$, the Jackknife mean, and s.d. for a critical degree of AMS ($AMS_c$). In **f** and **g**, comparisons are shown of individual and interactive effects (AMI:CTL × affected:unaffected limb) for muscle synergy similarity and synergy activation similarity for all 28 limbs (BIO: $n = 14$, AMI: $n = 7$, CTL: $n = 7$). Here paired one-tailed t-tests were used for BIO-A:AMI and BIO-C:CTL comparisons, unpaired two-tailed t-tests were used for the AMI:CTL comparison, and 2-way ANOVA was used for the interaction analysis (*$P < 0.05$, **$P < 0.01$). Where no significance is seen, a P value for the comparison is shown.

DoF motor control would be addressed as a change in either $\alpha_{PFDF}$ or $\alpha_{INEV}$, resulting in rectangle-type trajectories in $\alpha$-space. Therefore, we quantified 2-DoF motor controllability of each limb by computing the mean value of the diagonal components of traces in $\alpha$-space.

The 2-DoF motor controllability plotted against the degree of AMS (Fig. 3f) showed significant positive associations ($\tau = 0.58$, $P < 0.005$), and a relatively low $AMS_c$ was anticipated to preserve 95% of 2-DoF motor controllability for the residual limb ($AMS_c = 0.26$, $R^2 = 0.89$). Significant interactive effects (Fig. 3g, F = 12.22, $P < 0.005$) were found for 2-DoF motor controllability between amputation subgroup and limb category (AMI:CTL × affected:unaffected limb). Also, significant differences in 2-DoF motor controllability were found between CTL and BIO-C limbs ($t = 3.25$, $P < 0.01$) and between AMI and CTL limbs ($t = 3.42$, $P < 0.01$), whereas no significant difference was found between BIO-A and AMI limbs ($t = 0.60$, $P = 0.28$).

Together, our results suggest that AMS within residual muscles enhances the decoupling and stabilization of the motor behaviors for discrete ankle and subtalar joint movements, allowing robust, simultaneous 2-DoF motor control of the residual limbs without visual or other functional feedback. Further, our results imply that amputation procedures that actively preserve biological AMS may effectively preserve multi-DoF motor control after amputation.

**Spatiotemporal motor control under time constraints.** We investigated the impact of AMS within residual muscles on spatiotemporal motor control under time constraints through speed-accuracy motor tasks, also referred to as Fitts' law-type motor tasks[38–40]. We imposed time constraints, of 2.0, 1.5, 1.0, 0.8, or 0.5 s, within which subjects were asked to perform the 4 discrete ankle and subtalar movements (Fig. 4a). For each time interval setting, PF, DF, IN, and EV movement tasks were presented in randomized order. Spatiotemporal motor control performance was quantified from errors between target motor tasks and decoded motor intents in $\alpha$-space (Supplementary Fig. 4). An index of difficulty (ID) of this task was calculated for each time

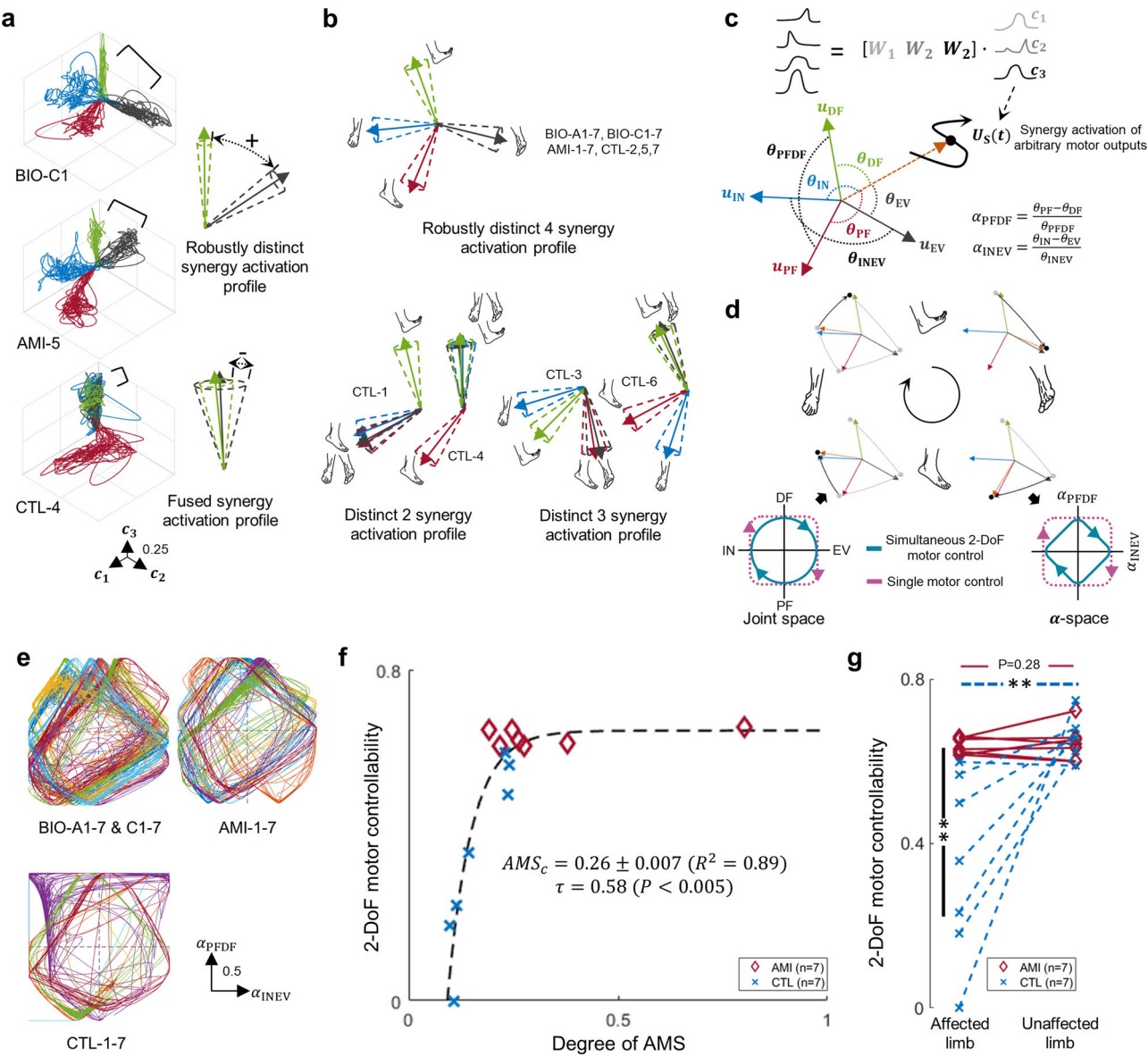

**Fig. 3 Multi-degree-of-freedom motor control.** In **a**, representative data are plotted that were collected during testing of the 4 discrete movements, and an illustration of robustly distinct and fused synergy activation profiles are shown. Here, the four different colors indicate the movements of plantarflexion (PF, red), dorsiflexion (DF, green), inversion (IN, blue), and eversion (EV, black). Each axis ($c_1$-$c_3$) indicates the activation of a muscle synergy ($W_1$-$W_3$). Agonist-antagonist Myoneural Interface (AMI); affected limbs of participants who underwent an AMI amputation (AMI-1-7); affected limbs of participants who underwent a Non-AMI control amputation (CTL-1-7); unaffected biologically-intact limbs (BIO-A1-7 and BIO-C1-7). In **b**, illustrations of the 4 discrete joint movements and the degree of decoupled motor behaviors are shown for synergy activation profiles of 14 subjects (AMI: $n = 7$, CTL: $n = 7$). Shown in **c** are diagrams of the $\alpha$-space transformation used to decode motor intention of an arbitrary motor output ($U_S = [c_1, c_2, c_3]$) from the average vectors of synergy activation profiles for the 4 discrete movements ($u_{PF}, u_{DF}, u_{IN}, u_{EV}$). Directionality of motor intentions in an ankle joint ($\alpha_{PFDF}$) and subtalar joint ($\alpha_{INEV}$) are computed based on angles ($\theta_{PF}, \theta_{DF}, \theta_{IN}, \theta_{EV}, \theta_{PFDF}$, and $\theta_{INEV}$) between $U_S$, $u_{PF}, u_{DF}, u_{IN}, u_{EV}$. Shown in **d** is the illustration of the draw-a-circle test and the relationship between joint space and $\alpha$-space. Simultaneous two-degrees-of-freedom (2-DoF) motor controllability of ankle and subtalar joints is defined as changes in both $\alpha_{PFDF}$ and $\alpha_{INEV}$ resulting in diamond-type trajectories in $\alpha$-space; single DoF motor control is defined as a change in either $\alpha_{PFDF}$ or $\alpha_{INEV}$. In **e**, motor intention trajectories are shown in $\alpha$-space for all 28 limbs (BIO-A & BIO-C: $n = 14$, AMI: $n = 7$, CTL: $n = 7$) during draw-a-circle trials. Distinct colors were superimposed indicating the different subjects. In **f**, the relationship between 2-DoF motor controllability and the degree of agonist-antagonist muscle strain (AMS) is shown in a combined analysis of affected AMI and CTL limbs ($n = 14$). Kendall's tau coefficients ($\tau$), P, $R^2$, and the Jackknife mean and s.d. for a critical degree of AMS ($AMS_c$) are reported. In **g**, comparison of individual and interactive effects (AMI:CTL × affected:unaffected limb) for 2-DoF motor controllability for all 28 limbs are shown (BIO: $n = 14$, AMI: $n = 7$, CTL: $n = 7$). Paired one-tailed $t$-tests were used for BIO-A:AMI and BIO-C:CTL comparisons, unpaired two-tailed $t$-tests for AMI:CTL comparisons, and 2-way ANOVA was used for the interaction analysis (**$P < 0.01$). Where no significance is seen, a P value for the comparison is shown.

constraint (2.0–0.5 s) as the logarithm of the inverse of the time constraint and scaled to range from 0 to 1 ($ID_{2.0}$-$ID_{0.5}$).

As visualized by 4 different colors in Fig. 4b, AMI-1 successfully performed the speed-accuracy motor tasks at all time constraint settings, producing 4 distinct synergy activations, one corresponding to each target motor task. Conversely, CTL-2 could successfully perform the speed-accuracy motor task only if allowed a longer time interval ($ID_{2.0}$ or $ID_{1.5}$) as evidenced by loss of the boundaries

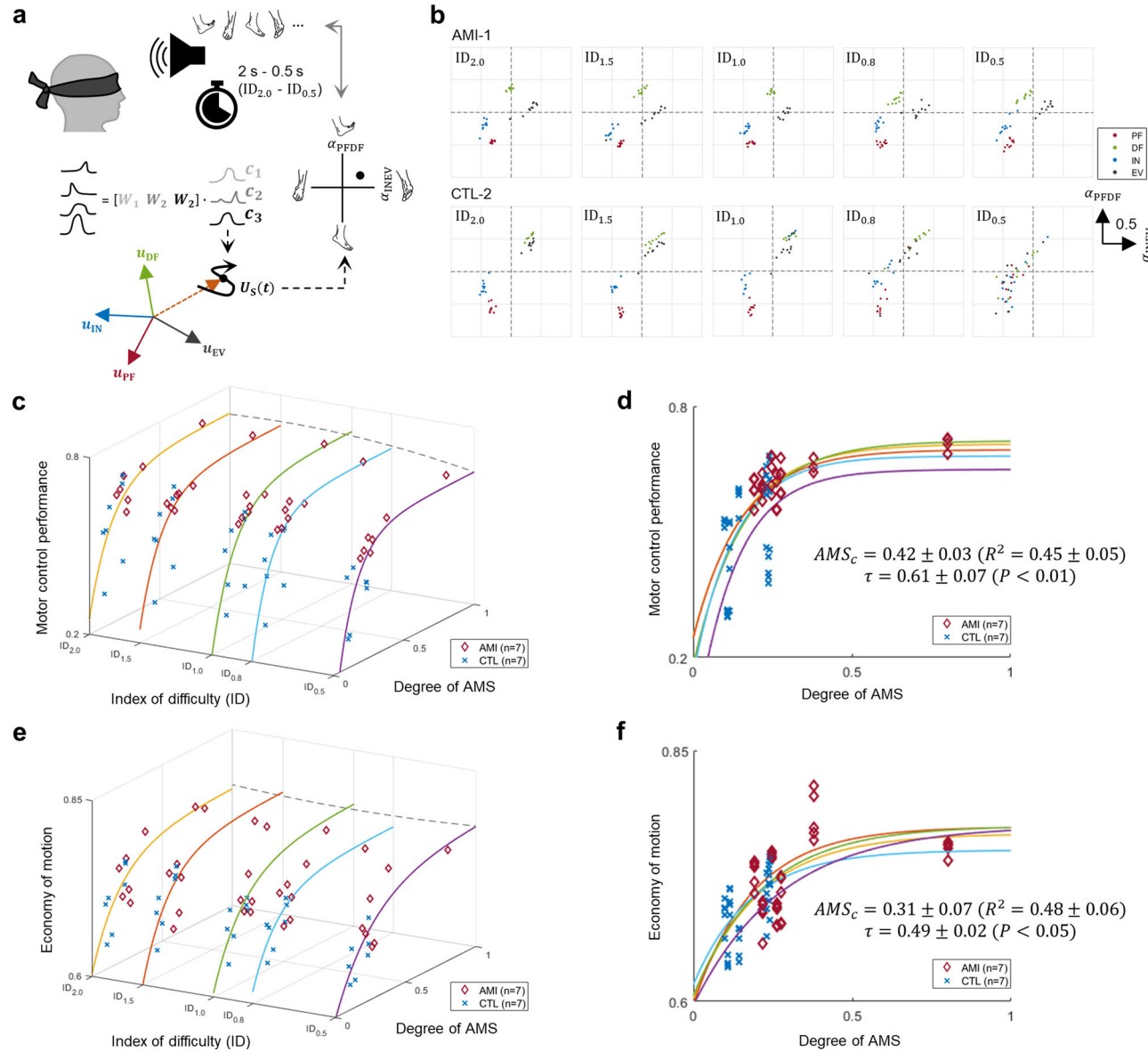

**Fig. 4 Spatiotemporal motor control under time constraints.** In **a**, an illustration is shown of a subject performing randomized, discrete ankle and subtalar joint motion tasks under varying time constraint (2–0.5 s), or index of difficulty ($ID_{2.0}$ – $ID_{0.5}$). To decode motor intentions of recorded motor outputs, first, synergy activations ($c_1$–$c_3$) are calculated using the muscle synergies ($W_1$–$W_3$) previously identified during 4 discrete motion testing of plantarflexion (PF), dorsiflexion (DF), inversion (IN), and eversion (EV). Finally, directionality of motor intentions in an ankle joint ($\alpha_{PFDF}$) and subtalar joint ($\alpha_{INEV}$) are computed based on the average vectors of synergy activation profiles for the 4 discrete movements $u_{PF}, u_{DF}, u_{IN}, u_{EV}$ and the synergy activation vector of recorded motor outputs ($U_S = [c_1, c_2, c_3]$). In **b**, motor control performance is shown for two representative subjects in $\alpha$-space ($\alpha_{PFDF}$ and $\alpha_{INEV}$). The Agonist-antagonist Myoneural Interface (AMI) subject maintained all 4 discrete movements up to the highest difficulty level with time constraint (0.5 s, $ID_{0.5}$). In contrast, the non-AMI control (CTL) subject started to lose boundaries at $ID_{0.8}$ and completely lost them at $ID_{0.5}$. Relationships are shown in **c** and **d** between the motor control performance and the degree of agonist-antagonist muscle strain (AMS) in a combined analysis of affected AMI and CTL limbs ($n = 14$) at all IDs. Further, relationships are shown in **e** and **f** between the economy of motion and the degree of AMS using a similar combined analysis ($n = 14$). Motor control performance and the economy of motion were evaluated by the error and straightness of motor control traces to the target motor tasks. Kendall's tau coefficients ($\tau$), P, $R^2$, and the Jackknife mean and s.d. of a critical degree of AMS ($AMS_c$) are reported.

between the 4 targets with increasing time constraint ($ID_{1.0}$-$ID_{0.5}$). Motor control performance plotted against the degree of AMS showed significant positive associations ($\tau = 0.61 \pm 0.07$, $P < 0.01$) across all IDs (Fig. 4c, d). We further investigated the economy of motion through trajectory straightness of the performed synergy activations during targeted movement tasks. Economy of motion responses plotted against the degree of AMS (Fig. 4e, f) showed significant positive associations against AMS ($\tau = 0.49 \pm 0.02$, $P < 0.05$).

Our results suggest that a higher degree of preserved AMS can provide a person with an amputation more sustained economy of motion, with minimized wasted movements, during spatiotemporal motor tasks. Conversely, a subject with limited AMS was less able to maintain efficacious motion and 'wandered' more during spatiotemporal motor control performance tasks shown as a low economy of motion. We speculate that the afferent feedback inherent in residual limb AMS may reduce 'trembling' of neuroprosthetic control in the absence of visual or other functional

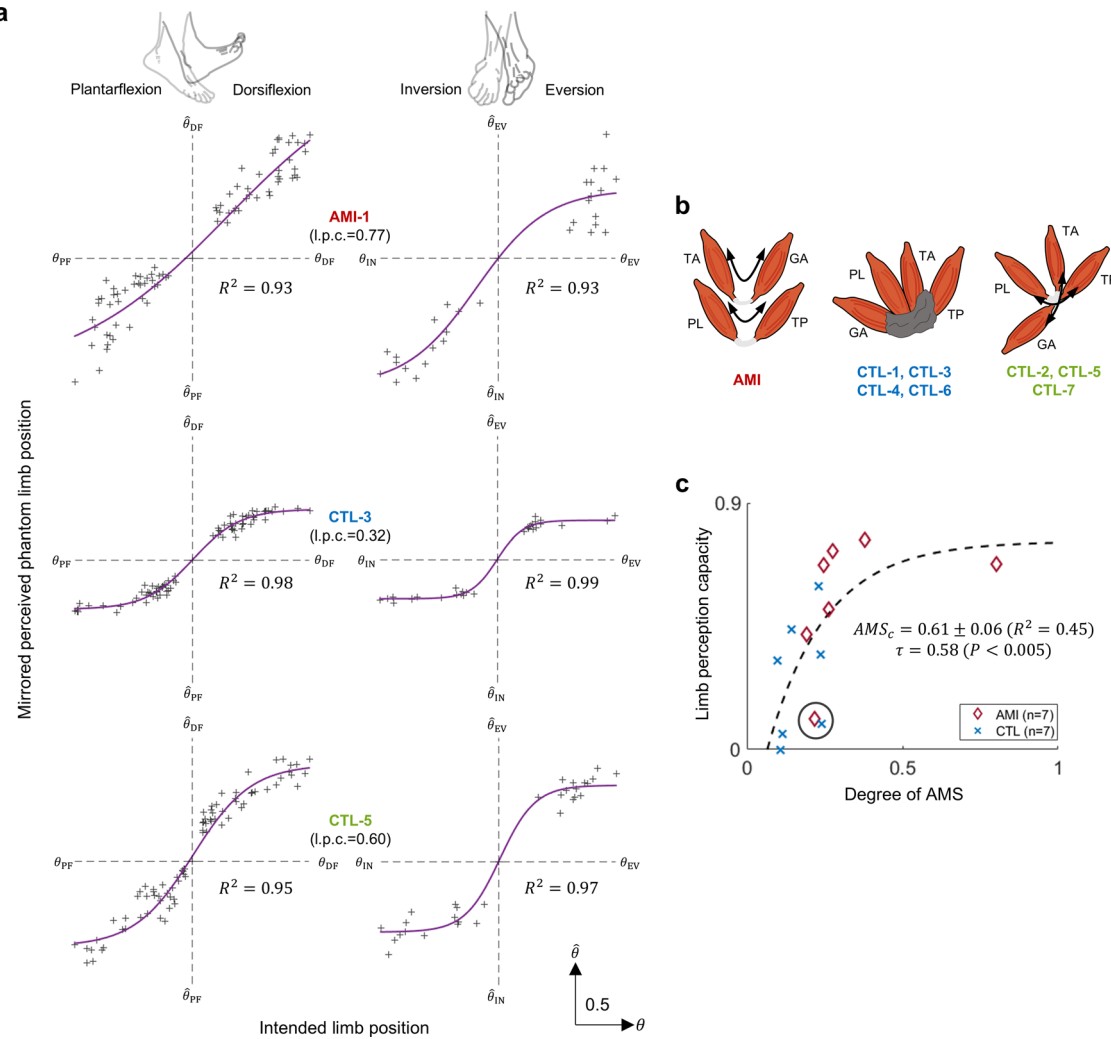

**Fig. 5 Subject-specific proprioceptive perception.** In **a**, relationships are shown between the mirrored perceived phantom limb position as measured from the biologically-intact limb without visual feedback and the intended phantom limb positions for three representative subjects (AMI-1, CTL-3, and CTL-5), who had respectively undergone Agonist-antagonist Myoneural Interface (AMI), traditional, and Ertl osteomyoplasty amputation procedures. Estimated limb perception capacity (l.p.c.) is reported. In **b**, schematic diagrams are shown of the anticipated residual limb structures, based on ultrasonography, highlighting the dynamic AMI construct excursion in 2-DoF for AMI, restricted motion in control subjects (CTL-1, CTL-3, CTL-4, and CTL-6), and 'joystick-like' residual muscle coupling in control subjects (CTL-2, CTL-5, and CTL-7). Size and positioning of elements are representative and not to scale. In **c**, limb perception capacity of the phantom limb is plotted against the degree of agonist-antagonist muscle strain (AMS) in a combined analysis of affected AMI and CTL limbs ($n = 14$). Kendall's tau correlation ($\tau$), $P$, $R^2$, and the Jackknife mean and s.d. of a critical degree of AMS ($AMS_c$) are reported. The limb perception capacity for AMI-2 and CTL-2 (encircled) deviated from the trend.

feedback to enable improved neuroprosthetic control, as will be further discussed in the Discussion section.

**Subject-specific proprioceptive limb perception.** Afferent signals from sensory receptors involved in proprioception[1] exert a strong influence on both motor control[41] and proprioceptive limb perception[4,41,42]. If the degree of residual AMS is a critical neurophysiological determinant underlying natural motor control preservation after limb amputation, we postulate that AMS should also contribute to preserving proprioceptive phantom limb perception. To explore this experimentally, we asked the subjects to move and vary their phantom foot positions to 25, 50, 75, and 100% of their DF or PF range of motion, and to 50 and 100% of their IN or EV range of motion, while mirroring phantom limb perception with their biologically intact foot. We assessed the intended phantom limb positions ($\theta$) from EMG as $\boldsymbol{U}_S \parallel \alpha_{PFDF}$ and $\boldsymbol{U}_S \parallel \alpha_{INEV}$ which indicate both direction and amplitude of the desired

movements (Fig. 5a). The measured mirrored perceived phantom limb position ($\hat{\theta}$) was accessed by goniometry and normalized by the range of motion of the BIO limb.

We estimated the values of limb perception capacity based on the psychometric function defined by plotting $\hat{\theta}$ against $\theta$ for each degree of freedom (Fig. 5a and Supplementary Fig. 5). Data for three subjects—one who had undergone the AMI (AMI-1), another a traditional (CTL-3), and the other an osteomyoplastic amputation[32,33] (CTL-5) - are shown in Fig. 5a. AMI-1 exemplified high limb perception capacity (0.77) and a high capability to vary her phantom foot position. In contrast, CTL-3 demonstrated lower limb perception capacity (0.32) and restricted capability to vary the position of his phantom foot. Notably, CTL-5, who had undergone an Ertl osteomyoplastic amputation[32,33], exhibited high limb perception capacity (0.60) and high capability to vary the position of his phantom foot. Consistently, when compared to the muscle dynamics presented

by AMI-1, CTL-3, and CTL-5, the ultrasound examination of CTL-5 revealed 2-DoF 'joystick-like' coupling between antagonistic muscle pairs that were distributed across the inferior aspect of his residual limb, as illustrated in the comparison shown in Fig. 5b.

Limb perception capacity plotted against the degree of AMS (Fig. 5c) showed a significant positive association ($\tau = 0.58$, $P < 0.005$) across all 14 residual limbs. However, the $AMS_c$ value was higher ($AMS_c = 0.61$, $R^2 = 0.45$) compared with the values of $AMS_c$ identified to preserve natural muscle synergy, synergy activation similarity, and 2-DoF motor controllability ($AMS_c$ ranged from 0.21-to-0.26). Our results suggest that subject-specific limb perception capacity is impacted by the degree of preserved residual muscle AMS after limb amputation. Our results further imply that a higher degree of preserved AMS within residual muscles may be required to preserve proprioceptive limb perception compared to the degree required to preserve biomimetic motor control. AMI-2 and CTL-2 (encircled, Fig. 5c), who deviated from the trend, reported no functional range of motion prior to their amputations during which time both experienced severe pain with attempted joint movements, as will be further discussed in the Discussion section.

**Supplemental clinical metrics**. We have gathered and analyzed clinical metrics including correlations between time since amputation and sensory-motor responses, maximum EMG values, average cross section of scar tissue, and average phantom limb score (Supplementary Fig. 6 and Supplementary Table 1). No significant correlations, positive or negative, were found between muscle synergy similarity (AMI: $P = 0.91$, CTL: $P = 0.42$), synergy activation similarity (AMI: $P = 0.15$, CTL: $P = 0.25$), 2-DoF motor controllability (AMI: $P = 0.33$, CTL: $P = 0.46$), or limb perception capacity (AMI: $P = 0.11$, CTL: $P = 0.89$) and time since amputation for the 14 residual limbs. Also, no significant differences between the AMI and CTL subjects were seen in the maximum EMG values recorded from the four target muscles (TA: $t = 1.41$, $P = 0.18$, TP: $t = -0.30$, $P = 0.77$, GA: $t = 1.49$, $P = 0.16$, PL: $t = -0.74$, $P = 0.48$), average scar tissue cross-sectional area ($t = 0.11$, $P = 0.91$), quantified with ultrasonography[43], or average phantom limb pain score ($t = 2.08$, $P = 0.06$).

## Discussion

In this study, we have accumulated evidence that, for the specific free space motor tasks performed, the degree of AMS within residual limb muscles postamputation is the neuromechanical determinant underlying the large variability observed in subject-specific motor control and perception. Our work supports the hypothesis that preservation of transtibial residual-limb AMS can restore sensorimotor capacity postamputation.

The single metric, AMS, a characteristic residual-limb structural feature, enabled the correlation of 6 types of sensorimotor responses from 14 transtibial amputee participants. Not unexpectedly, residual limb AMS and sensorimotor responses were individuated amongst the 14 participants, spanning a broad range of ages, times since amputation, surgical procedures, and etiologies. It was therefore remarkable that, despite this inherent intrasubject variability, residual limb AMS was found to impact motor control and perception as an exponential response. The gradual and monotonic improvements with the degree of preserved AMS underscore the importance of surgical amputation strategies like the AMI that preserve AMS.

Because muscle synergies and proprioceptive limb perception are organized at the central nervous system (CNS) level[26,28,44], the critical level of AMS identified in the exponential response implies CNS sensitivity to modified AMS in preserving motor control and perception. This CNS sensitivity to modified AMS may be critical to understanding and predicting outcomes of advanced surgical augmentation strategies for motor control and proprioceptive perception. The $AMS_c$ for muscle synergy, synergy activation similarity, and 2-DoF motor controllability range was 0.21–0.26, which predicts that about 21–26% AMS preservation will induce 95% preservation of natural motor control for discrete and multi-DoF movements. The finding that the $AMS_c$ for phantom limb perception capacity was relatively high (0.61) suggests a higher degree of afferent signaling from AMS may be involved in the context of proprioceptive memory modulation[9,45–47] compared to that required to preserve motor control. As a consequence of the difference in $AMS_c$ for motor control versus proprioceptive percepts, subjects with an AMS preservation value close to the 21–26% range demonstrated a natural level of motor control but yet exhibited a limited degree of proprioceptive percepts. Such was the case for study participants AMI-3, AMI-7, CTL-5, and CTL-7 with AMS values equal to 19.2, 26.4, 23.2, and 23.8%, respectively.

Muscle synergies in non-amputated humans reflect reflexive and afferent neural feedback in a manner dictated by musculotendon length changes and joint biomechanics[48]. To the best of our knowledge, the present work is the first to address the relationships between residual muscle biomechanics and muscle synergies in persons with major limb amputation. Together, our results suggest that AMS can provide a characteristic, readily ascertainable residual limb structural feature that can help to explain the variability in amputation outcomes and contribute together with other non-AMS factors including neuroplasticity, proprioceptive memory, neural signaling deficits, subject-specific perceptive responses, differences in scar tissue formation postamputation, and other biomechanical factors within the limb.

In this study, we address the beneficial outcomes of surgical paradigms that can actively preserve AMS during transtibial amputation. AMS preservation may also improve motor control and proprioceptive percepts for patients who undergo amputations at other anatomical levels. The AMI amputation procedure has now been conducted on over 30 patients at the transtibial, transfemoral, transradial and transhumeral levels to enable improved motor control and sensory perception in a broader population of amputee subjects[49]. When musculature distal to the amputation level is intact and viable, it may be harvested on a neurovascular leash during the amputation procedure, and AMS preservation can then be surgically implemented through muscle pair coupling and mechanical fixation of the construct to the surrounding fascia and muscle within the residuum[15,50]. If musculature distal to the amputation level is not viable, then AMS preservation may be implemented by constructing a native AMI using large vascularized muscle with TMR nerve reinnervation[15], or alternatively by constructing a regenerative AMI from reinnervated muscle grafts[15,51].

In some instances, it may not be feasible to surgically implement all agonist-antagonist muscle couplings to fully emulate intact-limb dynamics due to limited physical volume of the residuum skin envelope (e.g., in a transradial amputation). In this case, the AMI procedure may be applied to only major agonist-antagonist muscle pairs while other surgical strategies such as rPNI and TMR are applied to the remaining musculature sites and transected nerves. Thus, through an integration of AMI, rPNI, and TMR techniques, the amputation procedure can be designed to improve motor control and sensory perception for patient-specific functional limb restoration[15]. Another consideration is operative time; AMS preservation during amputation requires a somewhat extended operation time[25,52], and as such

may not be appropriate in some scenarios. In other cases, such as life-threatening emergencies, AMS may be excluded initially and introduced during a revision surgery, which is often performed to treat phantom limb pain or neuroma postamputation[20].

In this paper, we demonstrate the benefits of preserving AMS on motor control and sensory perception in free space using a synergy analysis that did not require a physical external prosthesis or a specific control system. In the EMG control of external powered prostheses, a critical challenge is the surface EMG electrodes needed for neuroprosthetic applications. Various research groups and companies are developing flexible, thin electrodes[5,53] for in-socket EMG recordings from residual muscles. We developed and investigated film-thick electrodes[54] and a neuroprosthesis[55], with which we are exploring AMI amputee subjects' capabilities for level ground walking and terrain adaptation. We are also integrating AMI constructs with an osseointegration implant, wherein a bone-anchored mechanical conduit can allow percutaneous passage of 16 implanted leads for electrical stimulation of, and recording from, an amputee subject's residuum muscles[56]. In this approach implanted EMG electrodes are placed onto each AMI muscle with each electrode's wire leads passing through the osseointegrated implant.

In this study, we present a platform that combines muscle synergy analysis with biophysical and biomedical sciences to clinically demonstrate the impact of AMS on physiological motor control after limb amputation in 14 amputee subjects. Naturalness of motor control is investigated using the extracted muscle synergies and their activation profiles. Our approach reveals that preserving a relatively low degree of AMS in the residuum (20–26% of that in a biologically intact limb) is effective in preserving natural motor control postamputation, though preserving limb perception capacity requires a higher degree (61%) of AMS preservation. Our findings suggest that AMS-amputation strategies are one way to enable more effective and biomimetic sensorimotor control postamputation. Previously, the reasons underlying motor control and proprioceptive perception postamputation were poorly understood. The unique circumstances of our study population and dataset, which comprises 28 AMI and CTL limbs - their post-amputation residual limbs and biologically intact limbs—allowed the discovery of a fundamental index. The present approach provides a predictive index of postamputation outcome, $AMS_c$, derived through a combination of mathematical, biomechanical, and clinical data. Consequently, our findings offer new, meaningful insights into motor and sensory perturbations by people living with major limb amputation. With further refinement, the concept of a critical degree of AMS may elucidate fundamental factors underlying clinical outcomes after amputation and inform future amputation paradigms and neuroprosthetic system designs.

For the motor tasks performed, our study provided evidence to support the hypothesis that the degree of AMS within the transtibial residuum is the neuromechanical determinant of subject-specific motor control and proprioceptive preservation. The main limitation of our study was its small population size— 14 transtibial amputees including 7 AMI and 7 non-AMI participants. We anticipate that future, larger studies will further elucidate the correlations established herein and provide further insight into how a multiplicity of factors such as age, time since amputation, body habitus, and amputation etiology may impact sensorimotor responses.

Emulating the natural biomechanics of muscle interaction to impart a high degree of AMS in residual limbs requires a sophisticated surgical design. Surgical considerations include physiological muscle tensions in mechanical couplings, balanced muscle force capacities, and minimizing the mechanical impedance to enable freely moving agonist-antagonist muscle dynamics. In practice, taking these multiple factors into account during the amputation procedure is challenging. In this study, the large variance in AMS across the AMI cohort underscores the need for further optimization of AMI amputation technique to enable a more consistent AMS clinical outcome. More broadly, an exciting area of future research would be the development of residual limb architectures that enable direct computer control of mechanoneural transduction[49]. To this end, basic research advances are necessary in physiological actuators such as biocompatible synthetic actuators[57], intramuscular sensors[58–60], and stimulators that can be implanted in series with residual muscle end organs.

The importance of mechanoneural transduction within sensory organs for afferent signaling is well known[2,4]. However, the motor control consequences of altered afferent signals by the macroscale reconfiguration of biomechanically-functional tissue architectures is not yet understood. Here we document the effects of modified afferent signaling from distinct functional tissue architectures of the transtibial residuum by revealing the neuromechanical determinant, and the sensitivity of its impact on the CNS, to subject-specific residual motor control and phantom limb perception. Toward the design of biomimetic neural interfaces, we wish to underscore the value of surgical techniques that create a residuum tissue structure that preserves natural neuromusculature and biomechanical function.

## Data availability
All data associated with this study are found in the main text and the Supplementary Materials. Source data underlying the main figures in the manuscript are available as Supplementary Data 1–3 and Supplementary Fig. 5. Individual participant data are not available due to limitations imposed by the informed consent form signed by the research team and study participants in advance of data collection.

## Code availability
Code for this study has been deposited in a DOI minting repository using Zenodo[61].

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

## Acknowledgements

We thank the participants for their dedication and time. We thank our clinical collaborators M.J. Carty, L. Berger and K. Clites. This work was supported by MIT Media Lab Consortia, fellowship support from the K. Lisa Yang Bionics Center (H.S.) and the Eunice Kennedy Shriver National Institute of Child Health and Human Development of the National Institutes of Health (NIH) under the grant R01HD097135 (H.M.H.). The content is solely the responsibility of the authors and does not necessarily represent the official views of the NIH.

## Author contributions

H.S. and H.M.H. contributed to study conceptualization and design. H.S. designed and implemented motor intent and phantom limb perception assessments. H.S., E.A.I., S.G.A., S.S.S., and L.E.F. performed data collection. H.S. analyzed residual motor control and phantom limb perception capacity with assistance from E.A.I.; H.S., E.A.I., S.G.A., and A.C.T. performed ultrasound data analysis. H.S. prepared the figures with assistance from E.A.I. and L.E.F.; H.S. wrote the manuscript with E.A.I., S.G.A., L.E.F., and H.M.H. All authors read and commented on the manuscript.

## Competing interests

The authors declare no competing interests.

## Additional information

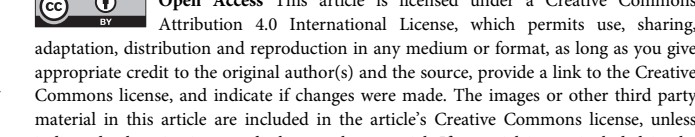

