## [Peer Review File · Communications Medicine]

Reviewers' comments:

Reviewer #1 (Remarks to the Author):

The manuscript describes measurement of muscle control and proprioception in 10 lower limb amputee subjects, specifically testing whether the type of amputation surgery affects the relative normalcy of subject control of the residual limb muscles and sensory perception. The idea for the study is appropriate and well-justified. The methodologies are relatively novel, largely as a result of the questions being asked. The results of the study are interesting and relatively easy to interpret. The overall finding supports the benefits of using agonist-antagonist myoneural interface surgery during lower limb amputations.

The biggest weakness of the manuscript is that it is very difficult to read. The manuscript is filled with abbreviations, jargon, and passive voice, reducing reading comprehension considerably. I strongly encourage the authors to read the guide to authors for Nature journals: <https://www.nature.com/nature-portfolio/for-authors/write>. Specifically:

"Nature journals prefer authors to write in the active voice ("we performed the experiment...") as experience has shown that readers find concepts and results to be conveyed more clearly if written directly."

"Many papers submitted for publication in a Nature journal contain unnecessary technical terminology, unreadable descriptions of the work that has been done, and convoluted figure legends."

"We ask authors to avoid jargon and acronyms where possible."

Please greatly reduce the number of abbreviations. Please try to predominately use active voice. Some passive voice is fine, but passive voice currently dominates the manuscript and makes it hard to read. Please reduce the use of jargon. For example, neuroergonomics is used in the manuscript incorrectly as it is not appropriate for what the authors are describing. The abstract, particular, is difficult to follow due to these issues.

Another weakness of the manuscript is a lack of measurement and discussion of scar tissue in the subjects. It is possible that differences in scar tissue could have a large impact on the differences across surgeries and subjects. The authors could have analyzed the ultrasound images to provide a quantitative assessment of scar tissue in a way that would have greatly added to the value of the manuscript. Alternatively, magnetic resonance imaging could have been used to get a better quantification of the scar tissue.

Minor weaknesses

In the statistics section, the authors write "the sample sizes used were similar to those reported in previous studies 22,23". That is not a particularly strong defense of the choice of number of subjects. Given the lack of a priori sample size estimation, it would be helpful if the authors could provide effect sizes for the main outcomes of the paper. That would greatly aid other researchers in the future attempting follow up studies.

It would be helpful to know if time since amputation had any correlation with the outcomes. There is a relatively small range included in the study but it would still provide value to examine it and comment on it in the discussion.

Additional comments or discussion of the differences in EMG amplitude between the different surgeries would add value to the manuscript.

Reviewer #2 (Remarks to the Author):

The authors describe their research findings comparing agonist-antagonist muscle strain in two different lower limb amputation procedures. They report that there is improved muscle pattern coherence with the specialized reconstruction procedure compared to the control. They also report phantom sensation appears to be correlated with the degree of motor control.

1. The findings are interesting but the primary limitation is small sample size and less uniform findings reported in sensory perception. This should be noted.
2. The discussion is somewhat speculative regarding the sensory perception and could be tightened by specifically discussing the findings and why there may be less uniformity in results.
3. The authors describe potential impact of this novel surgical procedure on prosthesis use and function but no data supports whether taking the time to preserve AMS will indeed improve prosthetic function or time of use.
4. The authors mention "frozen limb" phenomena in the discussion but it is not clear if any of the amputees had this condition and what the findings were from study procedures.

Reviewer #3 (Remarks to the Author):

The paper: "Agonist-antagonist muscle strain in the residual limb preserves motor control and perception after amputation" by Song et al, is continuing the research in the field of AMIs, from the same group, by expanding the patients' group and adding several new tests.

In the present manuscript authors show relationships between preserved agonist-antagonist muscle strain (AMS) within the residual limb and preserved motor control, together with the perception capacity investigations. Ten persons with unilateral transtibial amputations spanning a range of ages, etiologies were included.

The reviewer believes that this approach has a very high-potential broad implications in amputees' treatment. Reviewer also appreciates elegant tests and the use of the modern processing techniques as the synergies and neuroergonomic ones. Results are mainly convincing and, in the case of sensory part (Figure 5) well conveyed. I believe that the manuscript is valuable for the publication, but that an effort should be done from authors to make the figs and text more understandable.

While I do appreciate the study, its outcomes, numerosity of patients and tests, an effort must be placed in making the manuscript more accessible for the scientific readers that are not from the very same specific field. Figures clarity: Figures 2-4 are extremely complex and difficult to interpretate. Authors should place an effort in making them more informative also for the larger population of readers, especially from the medical field.

The suggestion could be using the graphical representations of the intended movements with respect to the computed parameters, and some simplified representation of the steps performed to arrive until synergies and other calculated coefficients from the raw data. This would help the readers to understand the steps performed, but also to other researchers to be able to replicate similar findings.

Figure 4 is particularly difficult to follow and should be done differently. It is not trivial to understand the alpha spaces and their physical meaning. Authors should help to the readers in understanding their general findings without the need to learn all the details of this very specific theory. One (of many) helpful examples of such “methodologically” explanatory figure can be found for instance in Gallego et al., 2018 Nat Comm. Figure 1 where they convey graphically the steps performed to calculate Neural Manifolds, which is mathematically the space reduction technique, as it is the synergy calculation.

Similar comments apply to the language used: e.g. “To assess the impact of AMS within residual muscles on neuroergonomics, defined as motor control performance during Fitts’s law-type motor tasks, time constraints of 160 2.0 s, 1.5 s, 1.0 s, 0.8 s, and 0.5 s were imposed for the performance of discrete ankle and subtalar movements....An index of difficulty (ID) was calculated as the logistic function of the inverse of).”

This should be rephrased into the language understandable to the wider population, not only the experts from the field.

MINOR:

“Several different surgical paradigms for higher-level amputation, such as a transfemoral amputation, may be designed based on the subject’s etiology.” —> It is not a clear statement, based on any explanation or citation, and therefore should be elaborated. In case of the missing evidence or citation this should be discussed as potential limitation of the present approach.

These results emphasize the imperative need to preserve antagonistic residual muscle dynamics to promote biomimetic neuroprosthetic control.

Please moderate the language since there are other ways to achieve the biomimetic control.

More discussion should be devoted to the placement of this technique within the range of other existing techniques (rPNI, TMR etc.) in discussion. Authors here propose in one short mention the use of osseointegration in combination with AMIs. Why and how? What is/are the envisioned future neurotechnological solutions for different patients’ categories?

Referee #1

The manuscript describes measurement of muscle control and proprioception in 10 lower limb amputee subjects, specifically testing whether the type of amputation surgery affects the relative normalcy of subject control of the residual limb muscles and sensory perception. The idea for the study is appropriate and well-justified. The methodologies are relatively novel, largely as a result of the questions being asked. The results of the study are interesting and relatively easy to interpret. The overall finding supports the benefits of using agonist-antagonist myoneural interface surgery during lower limb amputations.

The biggest weakness of the manuscript is that it is very difficult to read. The manuscript is filled with abbreviations, jargon, and passive voice, reducing reading comprehension considerably. I strongly encourage the authors to read the guide to authors for Nature journals: <https://www.nature.com/nature-portfolio/for-authors/write>. Specifically:

"Nature journals prefer authors to write in the active voice ("we performed the experiment...") as experience has shown that readers find concepts and results to be conveyed more clearly if written directly."

"Many papers submitted for publication in a Nature journal contain unnecessary technical terminology, unreadable descriptions of the work that has been done, and convoluted figure legends."

"We ask authors to avoid jargon and acronyms where possible."

Please greatly reduce the number of abbreviations. Please try to predominately use active voice. Some passive voice is fine, but passive voice currently dominates the manuscript and makes it hard to read. Please reduce the use of jargon. For example, neuroergonomics is used in the manuscript incorrectly as it is not appropriate for what the authors are describing. The abstract, particular, is difficult to follow due to these issues.

In response to this valid critique, the text and figure legends are fully edited to improve readability. An effort has been made to use the active voice, minimize the use of jargon, and reduce the number of abbreviations. The term 'neuroergonomics' is replaced with the phrase 'Spatiotemporal motor control under time constraints' in the Results (Lines 162-164), Figure 4, and the Methods (Line 445-447). The Abstract is fully revised as well. Given the extent of editing, not all manuscript revisions have been highlighted. However, we did highlight all changes that relate to a particular reviewer's concern.

Another weakness of the manuscript is a lack of measurement and discussion of scar tissue in the subjects. It is possible that differences in scar tissue could have a large impact on the differences across surgeries and subjects. The authors could have analyzed the ultrasound images to provide a quantitative assessment of scar tissue in a way that would have greatly added to the value of the manuscript. Alternatively, magnetic resonance imaging could have been used to get a better quantification of the scar tissue.

We agree and thank the Reviewer for this valuable suggestion. In response, we quantified cross-sectional scar tissue areas around the target residual limb muscles for each subject using our ultrasound images. No significant difference between the affected limbs of the AMI and CTL subjects was found, as is reported in the Results/Supplemental clinical metrics (Lines 220-221), and in Supplemental Fig. 6b.

Minor weaknesses

In the statistics section, the authors write "the sample sizes used were similar to those reported in previous studies 22,23". That is not a particularly strong defense of the choice of number of subjects. Given the lack of a priori sample size estimation, it would be helpful if the authors could provide effect sizes for the main outcomes of the paper. That would greatly aid other researchers in the future attempting follow up studies.

In response to this valid critique, we provide effect size for the main outcomes, calculated using Cohen's d as specified in the Methods/Statistics (Lines 481-482) and shown in Supplemental Table 2.

It would be helpful to know if time since amputation had any correlation with the outcomes. There is a relatively small range included in the study but it would still provide value to examine it and comment on it in the discussion.

We agree and thank the Reviewer this suggestion. We carefully examined correlations between the outcomes and time since amputation using Kendall's rank correlation. No significant correlations –either positive or negative– were found between time since amputation and muscle synergy similarity, synergy activation similarity, 2-DoF motor controllability, or limb perception capacity for the affected limbs of AMI and CTL subjects. These findings are reported in the Results/Supplemental clinical metrics (Lines 215-218). Also, the Discussion/Limitations and future directions (Lines 307-3010) acknowledges small study size as the major limitation of our study. We anticipate that future, larger studies will further establish the correlations established herein and provide further insight into how various factors such as age, time since amputation, body habitus, and amputation etiology may impact sensorimotor responses.

Additional comments or discussion of the differences in EMG amplitude between the different surgeries would add value to the manuscript.

In response to this comment, we reexamined our EMG data and reported maximum EMG amplitudes in the Results/Supplemental clinical metrics and in Supplemental Fig. 6a. Although EMG amplitude varied amongst the four target muscles (tibialis anterior, gastrocnemius, tibialis posterior, and peroneus longus), no significant difference between the affected limbs of the AMI and CTL subjects was found (Lines 219-220).

Referee #2

The authors describe their research findings comparing agonist-antagonist muscle strain in two different lower limb amputation procedures. They report that there is improved muscle pattern coherence with the specialized reconstruction procedure compared to the control. They also report phantom sensation appears to be correlated with the degree of motor control.

1. The findings are interesting but the primary limitation is small sample size and less uniform findings reported in sensory perception. This should be noted.

1a. We acknowledge the validity of the Reviewer's concern regarding our small sample size. In response, we increased our sample size by 40%, to fourteen subjects, in the revised manuscript. Our previous claims were unchanged with this larger sample size, and the statistical power was substantially improved for all analyses. Also, the Discussion/Limitations and future directions (Lines 307-310) acknowledges small study size as the major limitation of our study. We anticipate future, larger studies will further establish the correlations established herein and provide further insight into how various factors such as age, time since amputation, body habitus, and amputation etiology may impact sensorimotor responses.

1b. We agree with the Reviewer that our data shows weaker correlation and higher variability for sensory perception than motor control parameters. We note this and clarify our sensory perception findings in the Abstract (Lines 20-24) and Results (Lines 197-203 and 210-212).

2. The discussion is somewhat speculative regarding the sensory perception and could be tightened by specifically discussing the findings and why there may be less uniformity in results.

In response to this valid critique, we tightened up the Discussion/Relationships between neural indices, sensorimotor responses, and the degree of residual limb AMS. Further, we discuss our finding of less uniformity in sensory perception compared to motor control. Citing muscle synergy literature on afferent neural feedback in non-amputated persons [Bizzi & Cheung, 2013], we state that, to the best of our

knowledge, the present work is the first to address muscle synergies in amputated subjects, and as such afferent feedback adaptations are not yet fully understood (Lines 248-255). We more specifically outline possible explanations for the sensory perception variability that we found, including differing requirements for afferent signaling in proprioceptive memory modulation (AMI-2 and CTL-2, Lines 210-212; AMI-3, AMI-7, CTL-5, and CTL-7, lines 241-247), and aspects of the surgical amputation procedure (lines 197-203, Fig. 5b).

3. The authors describe potential impact of this novel surgical procedure on prosthesis use and function but no data supports whether taking the time to preserve AMS will indeed improve prosthetic function or time of use.

In response to this Reviewer concern, we retitled the relevant Discussion subheading as “Transitioning to neuroprosthetic control of a powered prosthesis” (Line 277). We clarify that the present work underscores the impact of AMS on the restoration of motor control and sensory perception for free-space performance testing (Lines 277-279). We note that it is beyond the scope of the present work to address whether preserving AMS can impact real-life prosthesis use and function.

Separately, the Reviewer correctly suggests that the AMI surgical procedure requires more operative time than a traditional amputation procedure. The average operative time for the AMI transtibial amputation is reported in the literature as 346 minutes for the first 3 transtibial patients [Clites et al, 2018] and, consistently, 340 minutes for the first 25 patients [Carty & Herr, 2021]. In response to the Reviewer’s comment, this information is now included in the Discussion subsection titled, “Surgical strategies” (Lines 272-274).

4. The authors mention "frozen limb" phenomena in the discussion but it is not clear if any of the amputees had this condition and what the findings were from study procedures.

We thank the Reviewer for noting that we neglected to mention which amputee subject reported the “frozen phantom limb” phenomena. We corrected our error in Supplemental Table 1 that CTL-1 reported “frozen phantom limb” and “ski boot sensation” during phantom limb sensation testing. Meanwhile, we have now revised the Discussion subsection titled, “Relationships between neural indices, sensorimotor responses, and degree of AMS in the residual” to better reflect our findings (Lines 241-247).

Referee #3

The paper: “Agonist-antagonist muscle strain in the residual limb preserves motor control and perception after amputation” by Song et al, is continuing the research in the field of AMIs, from the same group, by expanding the patients’ group and adding several new tests. In the present manuscript authors show relationships between preserved agonist-antagonist muscle strain (AMS) within the residual limb and preserved motor control, together with the perception capacity investigations. Ten persons with unilateral transtibial amputations spanning a range of ages, etiologies were included. The reviewer believes that this approach has a very high-potential broad implications in amputees’ treatment. Reviewer also appreciates elegant tests and the use of the modern processing techniques as the synergies and neuroergonomic ones. Results are mainly convincing and, in the case of sensory part (Figure 5) well conveyed.

I believe that the manuscript is valuable for the publication, but that an effort should be done from authors to make the figs and text more understandable. While I do appreciate the study, its outcomes, numerosity of patients and tests, an effort must be placed in making the manuscript more accessible for the scientific readers that are not from the very same specific field. Figures clarity: Figures 2-4 are extremely complex and difficult to interpretate. Authors should place an effort in making them more informative also for the larger population of readers, especially from the medical field. The suggestion could be using the graphical representations of the

intended movements with respect to the computed parameters, and some simplified representation of the steps performed to arrive until synergies and other calculated coefficients from the raw data. This would help the readers to understand the steps performed, but also to other researchers to be able to replicate similar findings. Figure 4 is particularly difficult to follow and should be done differently. It is not trivial to understand the alpha spaces and their physical meaning. Authors should help to the readers in understanding their general findings without the need to learn all the details of this very specific theory. One (of many) helpful examples of such “methodologically” explanatory figure can be found for instance in Gallego et al., 2018 Nat Comm. Figure 1 where they convey graphically the steps performed to calculate Neural Manifolds, which is mathematically the space reduction technique, as it is the synergy calculation.

We thank the Reviewer for this valid critique. In response, we thoroughly revised the Figures and Legends, and added five new conceptual illustrations (Fig. 2a, Fig. 3b-d, and Fig. 4a) to make our work easier to follow. Fig. 2a is a conceptual illustration of how motor control is executed by the activation of combinations of muscle synergies. Fig. 3c and Fig. 3d provide graphical representations of intended limb movements with respect to computed parameters. The synergy activation vectors representing the four discrete ankle and subtalar joint movements (u_{PF} , u_{DF} , u_{EV} , u_{IN}) are more clearly shown, as are the motions used during the draw-a-circle performance test. Simplified representations of the mathematical concepts and steps involved in the various manipulations and calculations, representations of the joint angles (α_{PDF} , α_{INEV}) and the relationship of joint angle vector components to an arbitrary motor output vector are illustrated in Fig. 3d. A conceptual illustration of how spatiotemporal motor control data were assessed under time constraint during discrete ankle and subtalar joint motion performance tasks is shown in Fig. 4a. We hope these Figure modifications make the work more accessible and easier to understand.

Similar comments apply to the language used: e.g. “To assess the impact of AMS within residual muscles on neuroergonomics, defined as motor control performance during Fitts’s law-type motor tasks, time constraints of 160 2.0 s, 1.5 s, 1.0 s, 0.8 s, and 0.5 s were imposed for the performance of discrete ankle and subtalar movements....An index of difficulty (ID) was calculated as the logistic function of the inverse of).” This should be rephrased into the language understandable to the wider population, not only the experts from the field.

In response to this concern, the manuscript was thoroughly edited to make our work more understandable by a broader readership. The term ‘neuroergonomics’ was replaced with more descriptive terminology ‘Spatiotemporal motor control under time constraint’ in Fig. 4 and the associated Methods. The objectionable sentences were revised (Lines 167-169). Please note that, due to the substantial nature of the rewrite, changes in readability are not specifically highlighted in the revised version whereas changes made in response to other Reviewer comments are highlighted.

MINOR:

“Several different surgical paradigms for higher-level amputation, such as a transfemoral amputation, may be designed based on the subject’s etiology.”—> It is not a clear statement, based on any explanation or citation, and therefore should be elaborated. In case of the missing evidence or citation this should be discussed as potential limitation of the present approach.

We apologize for being unclear in this text and citation. We revised the objectionable statement in the Discussion subsection titled, “Surgical strategies”, while providing explanations and citations (Lines 258-266).

These results emphasize the imperative need to preserve antagonistic residual muscle dynamics to promote biomimetic neuroprosthetic control. Please moderate the language since there are other ways to achieve the biomimetic control.

We agree and moderated the language, replacing the words ‘imperative need’ with the word ‘value’. The

sentence now states “Toward the design of biomimetic neural interfaces, *we wish to underscore the value of surgical techniques that create a tissue structure within the residuum that preserves natural neuromusculature and biomechanical function.* (Line 326-328).

More discussion should be devoted to the placement of this technique within the range of other existing techniques (rPNI, TMR etc.) in discussion. Authors here propose in one short mention the use of osseointegration in combination with AMIs. Why and how? What is/are the envisioned future neurotechnological solutions for different patients’ categories?

3a. In response to this concern, we have expanded the Discussion subsection titled, “Surgical strategies” to better describe how the AMI amputation fits with other existing surgical techniques (e.g. rPNI, TMR) (Lines 267-272). To this end, we added the citation titled “Reinventing Extremity Amputation in the Era of Functional Limb Restoration” by Herr et al.. To provide context and perspective, this critical paper was referenced as it was co-authored by several surgical colleagues who did pioneering work and who are actively engaged in TMR and rPNI procedures.

3b. In response to this concern, we clarified how osseointegration procedures may be combined with the AMI technique, and why such a combination may be useful for transitioning to real-life neuroprosthetic control (Lines 283-287).

REVIEWERS' COMMENTS:

Reviewer #1 (Remarks to the Author):

The authors have done an excellent job addressing all previous concerns.

Reviewer #2 (Remarks to the Author):

The authors have responded to reviewer comments and improved their manuscript. There are more patients and more accurate characterization of findings. The discussion is also written using more understandable language. The key findings are now focused on measured outcomes focused on surgical results.

Reviewer #3 (Remarks to the Author):

Authors mainly addressed my concerns. The work is sound and important. Unique minor advice is to get help for a final editing of the text, so to make a manuscript even more fluent for non-technical readers.

Reviewer #1 (Remarks to the Author):

The authors have done an excellent job addressing all previous concerns.

No comments or concerns to be addressed.

Reviewer #2 (Remarks to the Author):

The authors have responded to reviewer comments and improved their manuscript. There are more patients and more accurate characterization of findings. The discussion is also written using more understandable language. The key findings are now focused on measured outcomes focused on surgical results.

No comments or concerns to be addressed.

Reviewer #3 (Remarks to the Author):

Authors mainly addressed my concerns. The work is sound and important. Unique minor advice is to get help for a final editing of the text, so to make a manuscript even more fluent for non-technical readers.

We thank the reviewer for this suggestion. We further reduced the usage of abbreviations and improve structure of the overall manuscript to improve readability of our manuscript.